# Barriers to Medication Review Process Implementation—Cross-Sectional Study among Community Pharmacists in Jordan

**DOI:** 10.3390/healthcare10040651

**Published:** 2022-03-31

**Authors:** Mohammad Abu Assab, Hamza Alhamad, Inas Almazari, Bilyana Azzam, Hanadi Abu Assab

**Affiliations:** Department of Clinical Pharmacy, Faculty of Pharmacy, Zarqa University, Zarqa 13132, Jordan; halhamad@zu.edu.jo (H.A.); ialmazari@zu.edu.jo (I.A.); bolbol.alazzam16@yahoo.com (B.A.); abuassabhanadi@gmail.com (H.A.A.)

**Keywords:** medication review process, medication review service, pharmaceutical care, healthcare, barriers, community pharmacist, Jordan

## Abstract

The medication review process (MRP) is an extended, vital role of community pharmacists in improving health outcomes of medication use, yet it is neither systematically nor comprehensively provided bycommunity pharmacies in Jordan. This study aimed to identify the potential barriers hinderingMRP implementation bycommunity pharmacists in Jordan. A total of 550 community pharmacists electronically received a previously constructed and validated Arabic questionnaire explicitly developed to assess the current medication review practices and factors hindering the MRP, of whom 417 answered the questionnaire, giving a response rate of 75.8%. Among the investigated six categories’ seventeen barriers tothe implementation of the MRP, the highest rating was found for remuneration barriers (55.8%), followed by barriers related to regulations and patients, which scored 52.3% and 48.8%, respectively. Resource-related barriers were recognizedby 44.6% of participants, while qualifications and barriers related to physicians scored 42.9% and 41.8%, respectively. Although community pharmacists in Jordan are eager to extend their roles from traditional to more patient-centered ones, they encounter various barriers hinderingsuch development. Regulation adjustments accompanied by cost-effective remuneration and proper training are strong facilitators for community pharmacists to initiate the medication review service; make available the needed resources; and invest efforts, time, and money to operate it.

## 1. Introduction

Recently, there has been an urgent need to broaden community pharmacists’ roles in primary public health issues [1]. As a result, the range of services provided by community pharmacists has undergone rapid expansion, from traditional supply functions to more patient-focused services [2]. Community pharmacies comprise ideal sites for believable counseling that appeal to a large population segment. Furthermore, community pharmacists gain a unique understanding of the therapeutic management of the health needs of communities, operate underextended opening hours, and serve through daily interactions with patients, providing health education, immunization programs, disease awareness, and prevention initiatives [3]. The patient-centered roles of community pharmacists emphasize the achievement of optimal treatment results for the patient and the prevention ofhealth problems that patients may encounter because of the incorrect use of medicines [4]. The medication review process (MRP) is one of the crucial pharmaceutical care (PC) services that community pharmacists can provide, through which they perform a thorough review, evaluation, and discussion of patients’ medications to obtain the best treatment plan and to prevent any medication-related health problems [5,6].

The MRP is defined as a systematic and structured critical evaluation of a patient’s medicationto reach a treatment agreement, optimize medicationuse, minimize medication-related problems, improve health outcomes, and reduce waste [7,8]. According to the Pharmaceutical Care Network Europe (PCNE), a medication review (MR) is defined as a “structured evaluation of a patient’s medicines with the aim to optimize medicines use and improving health outcomes” [9]. Therefore, the MRP is one of the most critical aspects of the PC process, through which a complete assessment of the patient’s medications is carried out after determining the goals of the treatment and the patient’s needs and taking into account all of the factors that affect the selection of the appropriate treatment plan for the disease condition, thus developing a treatment plan that ensures the right medicines with the right doses are provided to the right patient [10].

Studies addressed the MRP’s positive outcomes on patients’ life, particularly the elderly, who receive more than one drug for a long time. That is, the ultimate goal of the MRP is to obtain the best possible treatment plan, improve the patient’s quality of life, reduce treatment costs, increase commitment to taking medication, and reduce potential medication-related side effects [10,11,12,13,14,15,16,17,18,19,20,21]. Therefore, the basis forconducting the process depends mainly on engaging indialogue with the patient to obtain all of the necessary information that mainly affects the treatment plan and the extent of the patient’s commitment to taking the medications included in the treatment plan completely and appropriately, thus ensuringthat the correct patient receives the correct medication at the correct dose [22].

A couple of MRP-related local studies were conducted; the findings showed that the service had a positive impact on patients’ health, that many problems related to the use of drugs were prevented from occurring when the MRP wasapplied, and that there wasacceptance and satisfaction from the patients about pharmacists performing the process due to the positive results on the patients’ health [23,24].

Although the MRP is widely adopted and studied globally [13,25], it is seldom provided bycommunity pharmacies in Jordan [23,24]. There is a lack of studies that address the potential barriers that hinderthe adoption of such a service bycommunity pharmacies. 

This study aimed to identify the potential barriers hinderingMRP implementation bycommunity pharmacists in Jordan, and it is expected to be the first in Jordan to do so. The outcomes of this study are valuable to supplement the decisionmakers (such as the Jordan Pharmacists Association (JPA)) with the key barriers and obstacles hinderingthe comprehensive implementation of the medication review service (MRS) by community pharmacists, thus enabling them to articulate plans and policies to bridge the gap between the current roles performed by community pharmacists and the needed future broader patient-centered roles.

## 2. Materials and Methods

Study design and settings: This was an observational, cross-sectional studyusing a pre-tested, validated questionnaire distributed to a sample of community pharmacies in Jordan. Data were collected over three months (from April to June 2021). Appendix A depicts a flowchart of the whole study process.

Study instrument: The study questionnaire was developed based on previous studies, a literature review [11,13,14,22,23,24], and research objectives. It was designed to be in Arabic to support participants’ responses, as Arabic is the official and mother-tongue language in Jordan. Six experts and five clinical pharmacists were individually approached to participate in the questionnaire validation process. Furthermore, a pilot study was performed to ensure that the questions were clear, understandable, and reflected the study’s objectives. Thirty community pharmacies were selected randomly, and the questionnaire with the study objectives was sent to them electronically, inviting them to participate. The pilot questionnaire electronic responses were collected over three weeks and tested for consistency and reliability before initiating the formal data collection process. Then, the updated, refined questionnaire was distributed electronically through social media (community pharmacies’ Whatsapp and Facebook groups), where pharmacists were invited to participate in the study questionnaire.

Sample size calculation and sampling strategy: According to JPA, there are currently around 3700 community pharmacies, with approximately 7200 community pharmacists. At a 95% confidence level and a 5% margin of error (significance α = 0.05) with a 50% response distribution, the minimum sample size was calculated to be 365 [26]. The questionnaire was distributed to a sample of 550 community pharmacies. 

Inclusion and exclusion criteria: The study participants were pharmacists working in community pharmacies in Jordan. All other pharmacy staff members and pharmacists working in settings other than community pharmacies were excluded.

Questionnaire measures: The electronic questionnaire contained three parts: the first was designed to obtain the socio-demographic characteristics of respondents, the second was dedicated to investigating the MRP implementation practices and behaviors performed by community pharmacists, and the third part was dedicated to identifying the barriers to MRP implementation that community pharmacists in Jordan recognize. 

Data analysis: The Statistical Package for Social Science (SPSS) software, version 23, was used to analyze the data generated from the study. The participants’ demographics arepresented using descriptive statistics (frequency/percentage). For the assessment of MRP practices and behavior questionnaire items (part two; 8 items), the Likert’s agreement five-point response scale (strongly disagree to strongly agree) was used. The Likert five-point response scale of frequency (never to always) was used to assess the barriers related to MRP implementation that community pharmacists in Jordan recognize(part three; 17 items). Descriptive statistics (frequency/percentage) were utilized to present responses on each scale, and missing responses were excluded from the calculation ofresponse percentages to the survey items. 

## 3. Results

A total of 550 community pharmacies from 12 governorates were approached to participate in the study, of which 417 answered the questionnaire, giving a response rate of 75.8%. 

### 3.1. The Socio-Demographic Characteristics of the Participants

An overview of the socio-demographic characteristics of the participants is presented in Table 1. The majority of participants were female (*n* = 321, 77.0%), hadan age ranging from 25 to less than 35 years (*n* = 212, 50.8%), had a BSc degree in pharmacy (*n* = 335, 80.3%), worked in independent pharmacies (*n* = 315, 75.5%), were from the northern governorates (*n* = 234, 56.1%), and had 3 years or less of working experience in community pharmacies (*n* = 240, 57.6%), and most of them were staff pharmacists (*n* = 317, 76.0%).

### 3.2. Reliability of Questionnaire

Questionnaire consistency and reliability were measured using Cronbach’s α test, where the overall internal consistency was excellent (Cronbach’s α = 0.920), with Cronbach’s α measures ranging from 0.901 to 0.950.

### 3.3. MRP Implementation Practices and Behaviors Performed by Community Pharmacists in Jordan

Table 2 shows the MRP practices and behaviors performed by community pharmacists in Jordan. The assessment revealed that 71.7% of them perform practices and behaviors related to the MRP, as 49.4% of them collect relevant data from patients or their caregivers. This is followed by the assessment of whether the prescribed medication is appropriate for the patient’s condition and whether the prescribed medications may cause health problems depending on the patient’s condition, at 48.0% and 45.8%, respectively. A total of41.7% of respondents stated that they are constantly improving their skills to conduct specialized MR. The assessment of whether the patient is satisfied with continuing the treatment and whether the patient still needs all of his/her medication was reported by 40.5% and 38.6% of respondents, respectively. Developing a plan that includes a follow-up with treating physicians and documenting the files of patients who underwent the MRP were performed by 36.7% and 33.3% of respondents, respectively. 

### 3.4. Barriers to MRP Implementation

Table 3 shows the reactions of Jordanian community pharmacists to 17 barriers to implementing the MRP, grouped into six categories: patient-related, physician-related, and resource-related barriers, in addition to regulation-related, qualification-related, and remuneration-related barriers. The highest rating was found for remuneration barriers (55.8%), followed by barriers related to regulations and patients, which scored 52.3% and 48.8%, respectively. Resource-related barriers were recognized by 44.6% of participants, while qualifications and barriers related to physicians scored 42.9% and 41.8%, respectively. 

## 4. Discussion

Although the MRP is widely adopted and studied globally, it is neither systematically nor comprehensively provided by community pharmacies in Jordan. Moreover, there is a lack of studies that address the potential barriers that hinder the adoption of such a service by community pharmacies. 

Data from a Jordanian study showed that the main barriers recognized to hinder the implementation of PC were pharmacists’ lack of pharmaceutical training (44.9%), lack of acceptability by physicians (43.4%), lack of supporting laws (42.0%), pharmacists’ lack of therapeutic knowledge and clinical problem-solving skills (39.4%), and pharmacists’ lack of communication skills (38.2%) [27]. Six categories of barriers were identified throughout this research. 

The majority of the community pharmacists believed that neither insurance nor patients were willing to pay for the MRS. Studies showed that conducting services such as MR would necessitate greater resources, such as additional staff, and, hence, necessitate an appropriate remuneration scheme. They also showed that adequate remuneration is an important facilitator for providing this type of service. Time and resource burdens are not problematic if the remuneration is adequate [11,14]. On the national level, given the current economic constraints, community pharmacists believed that paying for this service would be problematic and perhaps impede the adoption of such a service. Thus, healthcare benefits, notably safety, effectiveness, and cost management, must still be shown to policymakers to secure funding from health insurance companies, the government, and patients. Although adequate remuneration is necessary to allow initial investments, it is not the only facilitator of the provision of this kind of service and is generally not sufficient alone to put a new service into action in community pharmacies’ practice [11,14].

Community pharmacies’ legal conduct in Jordan is regulated through three central legislations, namely, the General Health Act, the Medication and Pharmacy Act, and the Jordan Pharmacists Association (JPA) Act. The provision of health and PC services bycommunity pharmacies, other than the preparation and dispensing of medications with related counseling to patients, is neither stated nor defined in these legislations; hence, community pharmacists must consider the regulation barriers to providing MR and any service. This study shows that only 22.7% of respondents believe that there is an approved reference for the service administration. In 2019, after a focused extensive effort by JPA, qualified community pharmacists were legally allowed to administer flu vaccination. JPA also exerts extra efforts in a similar direction to transform community pharmacists’ roles into more extended patient-centered ones. Such leaps are vital in facilitating PC services, including MR and medication therapy management (MTM).

The patient is the merit of the MRP, and his/her inclusion is the first step of its implementation. Thus, patient-related obstacles are at the heart of the process. The findings of this study show that less than one-quarter of the community pharmacists stated that patients accepted the pharmacist’s proposal to perform the MRP and that only 30.7% of them believed that patients provided them with enough information needed to perform the process. A study showed that patients’ refusal of the pharmacists’ proposal to be involved in the MRP could be attributed to several factors, including whether they think they do not need this service or whether they think that it is not the role of pharmacists. Some do not come to the pharmacy by themselves [11]. However, implementation of the MRP will be more straightforward if patients already have a good relationship with the pharmacist. Additional support, such as a broad media plan/program, to increase their awareness and perceptions toward the MRS and to highlight the benefits of this service on patients’ outcomes, particularly those with chronic disease and polypharmacy patients, will be the necessary beneficial facilitator. Concerning access to patient information, a study concluded that the lack of access to patient health information is an essential obstacle to MTM interventions. Access to electronic health records could help in this area [28]. However, this type of research is absent in Jordan, and further research is essential to determine the benefit of electronic health record access in community pharmacies. It will be a crucial future leap in community pharmacy-based patient-centered services. On a related front, almost 33% of community pharmacists in this study stated that they have documentation (manual or electronic records) of patients’ profiles to conduct the MRP. We can currently assume that those community pharmacists individually use their initiatives to develop and deliver services to their patients. In this instance, our focus throughout this study was primarily directed to identify the potential barriers hindering MRP implementation by community pharmacists in Jordan. Future research is needed by those who perform medication reviews to investigate their level of MRS implementation.

The majority of the community pharmacists who participated seemed unable to invest time and resources in developing new services. Concerning time, 71.5% of community pharmacists stated that they lack the time for the MRP; these findings are in line with studies on the MRP [11,14]. While the “lack of time” obstacle has frequently been noted in pharmacy practice research, this obstacle is caused by a lack of staff, directly tied to a lack of remuneration for this type of service. It is exacerbated by the pharmacy’s severe workload and administrative burden [29,30,31,32,33,34]. Therefore, time and resource constraints are not problematic if adequate remuneration is embraced [11,14]. Furthermore, pharmacists need to learn to delegate tasks better within their teams to make time in their already busy schedules. Efficient delegation requires good team communication within the pharmacy and special training for staff [11].

Our study shows that 48.7%, 41.7%, and 39.8% of pharmacists believe that the MRP needs a particular database, an appointment arrangement, and a dedicated place, respectively. Even though MRS in daily practice requires appropriate workflow management, such as physical space dedicated to conducting patient interviews and particular platforms, a dearth of research assesses the significance of this issue. A study investigated the barriers related to health information exchange that limit the ability to access and share accurate, complete, and timely medication data across the care spectrum and suggested several strategies to promote reliable data sharing across many systems with integrated data sharing infrastructures (e.g., plans, electronic health records, pharmacy systems, retail systems, and personal applications). Emerging technologies, such as digital therapies, pharmacogenomics, precision medicine, and artificial intelligence-driven services, will require such infrastructures. In addition, the rising body of evidence suggests that embracing a paradigm change in data ownership and management, in which individuals play an increasingly central role in accessing, owning, and re-sharing their data across their lives, can lead to revolutionary progress [35].

Commitment, support, and engagement from community pharmacies’ owners and top management are crucial for the successful implementation of any service. Unfortunately, only 37.4% of the study respondents reported that they gained the owners/pharmacy managers’ support for the MRP. Fortunately, only 18.9% of the participants believed that the MRP was a waste of time. This issue implies that community pharmacists in Jordan address the importance of the MRP and that most of them are eager to carry it out formally in their pharmacies.

In alignment with other studies [11,23,36], this study’s findings reveal that a lack of adequate training related to the MRP isa significant barrier, as only 23.5% of participating community pharmacists had sufficient training and practical application on how to conduct the service. In addition, only 30.2% of respondents stated that the MRP was taught at the undergraduate level, and 33.6% believed that there are no qualified ortrained pharmacists to carry out this process. One study from Jordan reported that the provision of PC is limited and that the lack of PC training is a significant barrier for PC implementation [36]. Training appears to be an essential facilitator of the practice change process in community pharmacies [37]. Jordanian community pharmacists are seriously concerned about training and clinical skills, as are pharmacists in many other countries regarding patient-centered care and health promotion [27]. The training goals for the implementation of the MRP must include essential high-level pharmacotherapy (tailored to primary care practice), pharmacoeconomics, and services, as well as management skills and specific medicine review abilities (for example, selecting patients and who has priority; conducting data analysis to identify eligible patients; conducting patient interviews; and writing reports). Computer skills are also required (data management and outcome monitoring) [38,39]. Policymakers should keep training in mind when planning strategies to adopt modern programs or services, and academics should incorporate these features into pharmacist training. Changes in undergraduate pharmacy curricula are required to guarantee that students learn more about patient-focused issues and the relevant parts of information management and technology, behavioral sciences, communication, and health problem solving. Doing so will ensure that the students gain the necessary information and skills for patient care practice. It is critical to introduce postgraduate programs that place a greater emphasis on patient-centered teaching and training. Wherever possible, pharmacy schools should be administratively positioned so that the combined training of health professionals is possible. Pharmacy owners should also be aware of the need to involve their entire workforce in the implementation process, even if the service is seemingly supplied solely by the pharmacist, and should engage staff members in the planning and goal-setting procedures.

Establishing a good relationship with the treating physician is essential for the success of the MRP, as this study reveals that only 22.7% of community pharmacists believe that the patients’ physician will accept the results and outcomes of the MRP and that 48.9% of them believe that a good relationship is required for successful implementation of the service. Despite the many benefits of establishing and maintaining positive relationships between community pharmacists and physicians, doing so is not always straightforward. The need for collaborative practice agreements has been documented in the pharmacy literature worldwide [40,41,42,43]. Another critical study about physicians’ attitudes toward pharmacist-provided MTM treatments emphasized the importance of direct joint pharmacist–physician coordination of care [44]. However, a study showed that physicians in Jordan accept the pharmacist’s traditional role. They are, however, apprehensive of the adoption of additional clinical responsibilities [45]. According to a study, the development of inter-professional workshops in collaboration with various healthcare associations might be examined to allow pharmacists and physicians the opportunity to meet face to face and discuss shared objectives [46]. In addition, changes to reimbursement models and infrastructures, such as province-wide drug information systems and electronic health records, may be needed to realize the full benefit of collaborative practice between pharmacists and physicians to achieve optimal quality and outcomes of patient care [47]. Thus, we need to develop strategies and interventions to encourage collaboration with a more profound knowledge of the physician’s connection. The most vital strategies to implement are to encourage positive attitudes and the perception of helpfulness, and for health administrators and professionals to take advantage of new changes enforced by the health system as an opportunity to initiate collaboration, as well as promoting face-to-face relationship development to overcome prejudices, enable teamwork initiation and development, and designate coordinators responsibilities. Future studies are required to assess the efficacy of these tactics, as well as the further assessment of physicians’ perceptions toward the MRP in Jordan. 

Although the findings of our study in Jordan are almost comparable to those of other studies conducted in other countries worldwide, we needed to conduct this study to determine where we are currently, what our ground base is regarding the barriers to MRP implementation, and where to start building the capacity for the service in Jordan.

Comprehensively, a multi-stakeholder engagement approach that involves pharmacy colleges, professional associations (pharmacists and physicians), health policymakers, and health insurance is essential for the development and practice implementation of the MRS.

### 4.1. Study Strengths and Limitations

This study is expected to be the first or among the few that addressed the potential barriers that hinder the adoption of MRS by community pharmacies in Jordan. Thus, this study’s outcomes are valuable foundations to supplement the decisionmakers, such as JPA, with the key barriers and obstacles hindering the comprehensive implementation of the service by community pharmacists, consequently enabling them to articulate plans and policies to bridge the gap between the current roles performed by community pharmacists and the needed future broader patient-centered roles.

This research was conducted during the ongoing COVID-19 pandemic, explaining the use of an electronically distributed questionnaire as a data collection tool, giving rise to a sort of response bias toward youth community pharmacists with higher interaction; follow-up; and familiarity with electronic, digital, and social media platforms compared to older counterparts. 

For future research, we suggest, if possible, establishing a list of all licensed community pharmacies in Jordan with their geographic distribution and using both electronically and personally distributed surveys to ensure the representation of all community pharmacists’ age groups and governorates where pharmacies are located.

### 4.2. Implications for Future Research

The investigation of the level of MRS implementation by community pharmacists who perform medication reviews and the proposal of a feasible business model for community pharmacy-based MRSs are areas of future research interest to whoever is concerned with optimizing health and economic outcomes of medications and other therapies for the population of Jordan.

## 5. Conclusions

Our study identified six categories of barriers that currently hinder MRP implementation by community pharmacists in Jordan. The findings from this study pave the way for health policymakers and decisionmakers, such as JPA and the Ministry of Health, to develop plans and policies to effectively and efficiently take advantage of community pharmacies’ capacity to serve patients and the healthcare system as a whole, explicitly removing the constraints that hinder community pharmacists’ adoption and execution of more prominent patient-centered roles. Regulation adjustments to allow community pharmacists to exert the MRP, accompanied by a cost-effective remuneration scheme and proper training, are solid motivators for community pharmacies to initiate the service; make needed resources available; and invest efforts, time, and money to operate it. 

## Figures and Tables

**Table 1 healthcare-10-00651-t001:** Socio-demographic characteristics of the participants (N = 417).

Characteristic		N	%
Gender	Female	321	77.0
	Male	92	22.0
	Missing data	4	1.0
Age	25 to less than 35 years	212	50.8
	35 to less than 45 years	45	10.8
	More than 45 years	34	8.2
	Less than 25 years	123	29.5
	Missing data	3	0.7
Qualification	Bachelor (Doctor of Pharmacy)	27	6.5
	Bachelor (pharmacy)	335	80.3
	Ph.D.	2	0.5
	Master’s degree	40	9.6
	Missing data	13	3.1
The governorate in which the pharmacy is located	Central governorates	163	39.1
	Southern governorates	17	4.1
	Northern governorates	234	56.1
	Missing data	3	0.7
Pharmacy Ownership	Chain pharmacy	89	21.4
	Independent pharmacy	315	75.5
	Missing data	13	3.1
Job Title	Pharmacist owner	20	4.8
	Owner and responsible pharmacist	50	12.0
	Employee pharmacist	317	76.0
	Other	18	4.2
	Missing data	13	3.0
Practical experience in community pharmacies	More than 10 years	51	12.2
	3-6 years	69	16.5
	6-10 years	51	12.2
	Less than 3 years	240	57.6
	Missing data	6	1.5

**Table 2 healthcare-10-00651-t002:** Medication review process (MRP) practices and behaviors performed by community pharmacists in Jordan (N = 417).

Rank	Practice/Behavior	N	%
1	We collect relevant information from patients or their caregivers.	206	49.4
2	We assess whether the prescribed medication is appropriate for the patient’s condition.	200	48.0
3	We assess whether the prescribed medications may cause health problems depending on the patient’s condition.	191	45.8
4	We are constantly improving our skills to conduct specialized medication reviews.	174	41.7
5	We assess whether the patient is satisfied with continuing the treatment.	169	40.5
6	We assess whether the patient still needs all of his/her medications.	161	38.6
7	We develop a plan that includes follow-up with the patient’s treating physician.	153	36.7
8	We have documentation (manual or electronic records) of patients’ profiles to conducting the medication review process.	139	33.3

**Table 3 healthcare-10-00651-t003:** Barriers to MRP implementation bycommunity pharmacists in Jordan (N = 417).

Rank	Barrier Category (%)	Barrier Items	N	%
1	Remuneration related (55.8%)	Insurers are ready to cover a fee for the pharmacist to perform the medication review process for their insured patients.	86	20.6
		Patients are willing to pay the pharmacist for the medication review process.	87	20.9
		The pharmacy provides a financial incentive for pharmacists who conduct medication re-views	130	31.2
2	Regulation related (52.3%)	There is an approved reference and specific steps for the pharmacist to carry out the medica-tion review process.	95	22.8
3	Patient related (48.8%)	Patients accept the medication review process by the pharmacist.	82	19.7
		The patients provide the pharmacist with the information needed to perform the medication review process.	128	30.7
4	Resource related (44.6%)	We can provide enough time to do the medication review process.	119	28.5
		Doing a medication review requires the availability of specialized databases.	203	48.7
		Implementing the medication review process requires the appointment of a pharmacist dedi-cated to this purpose.	174	41.7
		The medication review process requires a designated place in the pharmacy.	166	39.8
		The pharmacy management supports the medication review process.	156	37.4
		Performing medication reviews is a waste of a pharmacist’s time.	79	18.9
5	Qualification related (42.9%)	Pharmacists have sufficient training and practical applications to conduct the medication re-view process.	98	23.5
		The medication review process is taught during the undergraduate level.	126	30.2
		Qualified and trained pharmacists are available to carry out the medication review process.	140	33.6
6	Physician related (41.8%)	The treating physician accepts the results and outcomes of the medication review process.	95	22.8
		The success of the medication review process requires good relationships with the treating physicians.	204	48.9

## Data Availability

The data presented in this study are available on request from the corresponding author.

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
