# Peer review of "Barriers to Medication Review Process Implementation—Cross-Sectional Study among Community Pharmacists in Jordan"

_healthcare, 2022, doi:10.3390/healthcare10040651_

Round 1
Reviewer 1 Report
In the manuscript “Barriers Towards Medication Review Process Implementation-Cross-Sectional Study Among Community Pharmacists in Jordan”, the authors use survey to address the potential barriers confronting Medication Review Process (MPR) implementation among community pharmacists in Jordan. Their analysis shows that the rates of the barriers are remuneration (55.8%), regulations (52.3%), patient-related (48.8%), resources-related (44.6%), qualification-related (42.9%), and physicians-related (41.8%), respectively. So, the authors conclude that the top three rating barriers are strong facilitators for the community pharmacists to initiate the medication review service in Jordan.
The data and findings of this study are useful information for some countries or regions to enhance their medication review service.
Here are some suggestions that might help improve the manuscript:
1. The abbreviation “MRS” means medication review service?
(Page 2, Line 77;
Page 7, line 198, and line 234;
Page8, Line 254;
Page 9, line 346 and 352.)
2. The section of the materials and methods: it might be easy to follow to represent the whole study process with a flow chart.
3. It could be better to summary the results with a circle graph (parts of whole graph).
4. Is there any suggestions about the data collection such as conducts for improvement for future research?
Reviewer 2 Report
The authors described in details the barriers confronting Medication Review Process (MRP) implementation among community pharmacists in Jordan. Although the study was conducted in Jordan, the results might be of interest for other nations. The study identifies the advantages and obstacles with which the pharmacists may face worldwide. The authors emphasize also the need for the implementation of the aspect of MRP in the academic training, which is of great importance. The manuscript is well-written and interesting. My only comment is that in lines 47, 150, 151, 158 the abbreviation ‘MR’ was not explained and it seems as it stands for ‘medications review process’. The explanation appears as a table capture (line 164).
Reviewer 3 Report
Comments for: Barriers Towards Medication Review Process
Implementation-Cross-Sectional Study Among Community Pharmacists in Jordan
Lines 70-72 are missing references to support the theses.
The authors state that they used their own authorship, based on the scientific literature, a validated questionnaire. Please provide references on the basis of which the research questionnaire was created.
The authors report that 417 pharmacists answered the questionnaire, giving a response rate of 75.8%. What about missing data?
How many pharmacists were excluded from the study based on the inclusion and exclusion criteria?
In table no.3, I propose to additionally introduce the number N, similarly to table no.2.
I suggest supplementing the discussion with the strengths and limitation of the study.
The conclusions should be rewritten, they do not correspond with the obtained results.
Reviewer 4 Report
The implementation of medication review in community pharmacy is a worldwide challenge. The present manuscript is centred in Jordan, but the problems found by the authors do not differ from any other country.
I think that the main problem is that these problems exist since always, but little progress has been done to solve them.
Some thinks I would like to comment about the present work:
- From what I can read in the manuscript I conclude that community pharmacists do not have access to the clinical record of the patient? Is that so? This is an important issue and one of these chronic problems.
- The pharmacists that performed reviews, did they have any system for recording their interventions? As the authors indicate in the discussion, in order to demonstrate the value of MRP it is necessary to have data about the work done by the pharmacist.
- Discussion: As one of the challenges found is the training of pharmacist, the authors indicate the need to include these issues in curriculum of studies of pharmacy. I can say that in many Universities we have these aspects included, but if after the graduation the pharmacist cannot apply them it his/hers everyday practice, with time these skills are forgotten.
Author Response
Please see the attachment

This manuscript is a resubmission of an earlier submission. The following is a list of the peer review reports and author responses from that submission.